# Convoluted micellar morphological transitions driven by tailorable mesogenic ordering effect from discotic mesogen-containing block copolymer

Huanzhi Yang[1], Yunjun Luo[1,2], Bixin Jin[1] ✉, Shumeng Chi[1,3] & Xiaoyu Li [1,2,3] ✉

Solution-state self-assemblies of block copolymers to form nanostructures are tremendously attractive for their tailorable morphologies and functionalities. While incorporating moieties with strong ordering effects may introduce highly orientational control over the molecular packing and dictate assembly behaviors, subtle and delicate driving forces can yield slower kinetics to reveal manifold metastable morphologies. Herein, we report the unusually convoluted self-assembly behaviors of a liquid crystalline block copolymer bearing triphenylene discotic mesogens. They undergo unusual multiple morphological transitions spontaneously, driven by their intrinsic subtle liquid crystalline ordering effect. Meanwhile, liquid crystalline orderedness can also be built very quickly by doping the mesogens with small-molecule dopants, and the morphological transitions are dramatically accelerated and various exotic micelles are produced. Surprisingly, with high doping levels, the self-assembly mechanism of this block copolymer is completely changed from intramolecular chain shuffling and rearrangement to nucleation-growth mode, based on which self-seeding experiments can be conducted to produce highly uniform fibrils.

Self-assembly behaviors are ubiquitous in nature, in which a random batch of subunits spontaneously forms a specifically organized structure to minimize the systematic energy through specific, local interactions among the subunits without external directions[1–3]. Particularly, many self-assembly processes in natural systems are achieved in a very subtle approach so that the subunits are organized in a quasi-equilibrium manner[4,5]. The convoluted interplay and delicate balance between different interactions make these assemblies very sensitive to external stimuli and can go through multiple metastable intermediate states to reach the final thermodynamic-stable state[6,7]. For example, endosomes can deform under even subtle physiological stimuli into various assembly morphologies to realize different biological functions, such as thin tubules, giant vesicles, endosomal carrier vesicles, and multivesicular bodies[8,9].

To comprehensively understand those mysterious natural assembly behaviors, mimicking such behaviors with artificial systems is vital[10]. Especially, block copolymers (BCPs) with pendant groups can combine both functional moieties and linear chain-like molecular structures, providing excellent models to investigate how molecular-level changes can influence the macromolecular organization and the assembled morphologies[11]. However, in most of these cases of BCP self-assembly, the driving forces are overwhelming and dictate the micellar morphologies, such as the UV irradiation-induced morphological switch between toroids and barrels from azobenzene-

[1]School of Materials Science and Engineering. Beijing Institute of Technology, 100081 Beijing, China. [2]Key Laboratory of High Energy Density Materials, MOE. Beijing Institute of Technology, 100081 Beijing, China. [3]Experimental Center of Advanced Materials, Beijing Institute of Technology, 100081 Beijing, China. ✉ e-mail: bixinjin@bit.edu.cn; xiaoyuli@bit.edu.cn

containing BCPs[12], the crystallization-induced transformation of emulsion droplets to form faceted vesicles[13], pH-induced inversions of bilayered vesicles[14], osmatic pressure-induced deformations of vesicles to form stomatocytes[15]. Unfortunately, with such strong driving forces, the subtle transitions are often covered up and largely neglected, making it very hard to emulate the intricacy, homogeneity, and versatility of natural systems. The construction of delicately balanced systems with the convoluted interplay of different subtle interactions is still very challenging but also attractive from the point of view of fundamental research[16,17].

Herein, we synthesized a diblock copolymer with a liquid crystalline (LC) block containing 2,3,6,7,10,11-hexakis(hexyloxy)

triphenylene (HAT) discotic mesogens, and closely studied its convoluted LC ordering-driven self-assembly behaviors in solution. Through dispersion in selective solvent *via* heating, the resultant micellar aggregates would go through multiple morphologies spontaneously from spherical to worm-like, tubular, vesicular micelles, and finally thin fibrillar shapes over several months. These transitions were driven by the intrinsic slowly-building LC orderedness from the discotic mesogens (Fig. 1b upper). More interestingly, it was found that when 2,4,7-trinitrofluorenone (TNF) was introduced to dope the mesogens, their organization would be dramatically accelerated and the orderedness would also be greatly enhanced to quickly produce various exotic micellar morphologies such as elongated vesicles, and

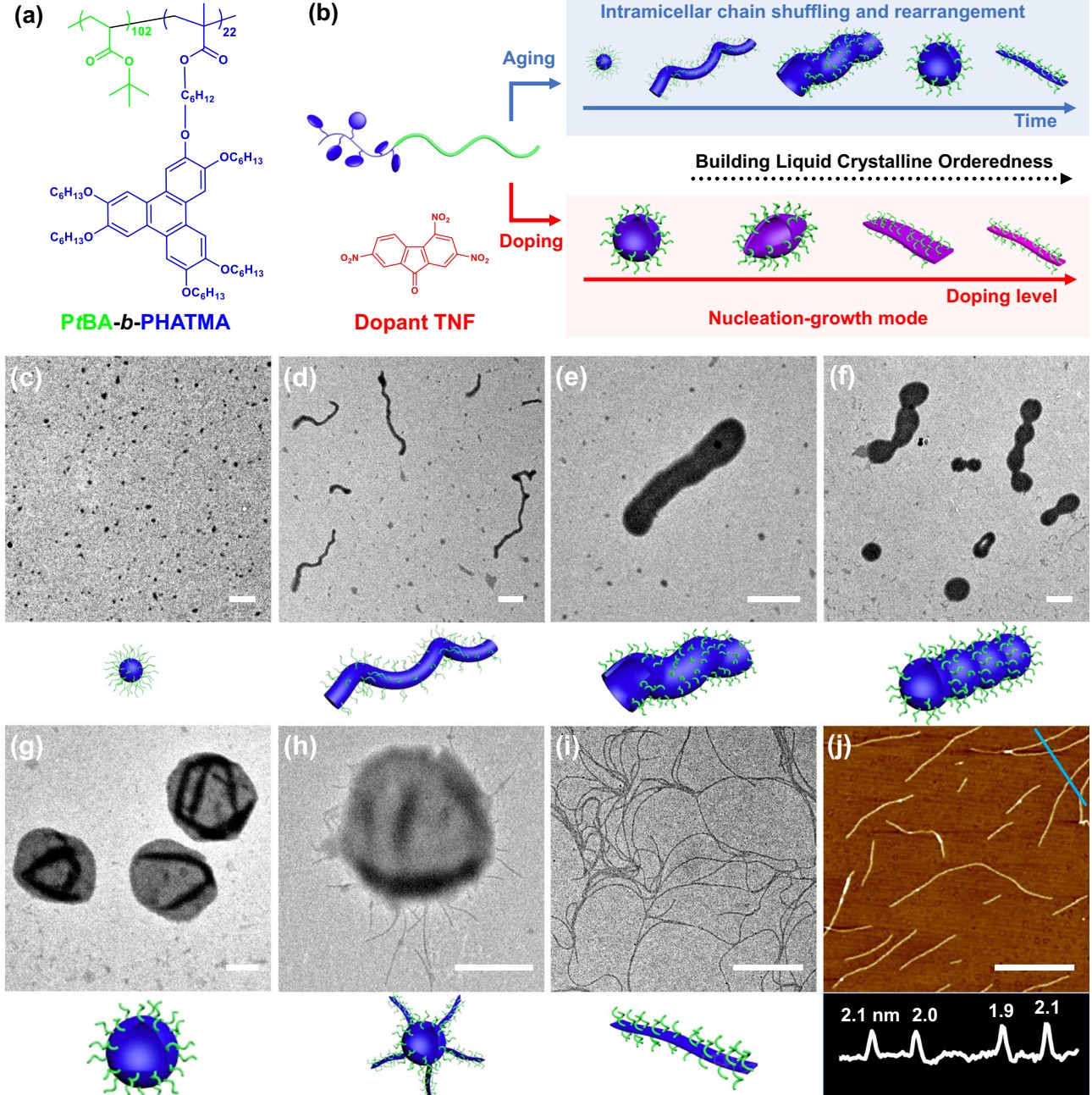

**Fig. 1 | Convoluted morphological transitions. a** Chemical structures of P*t*BA-*b*-PHATMA and TNF. **b** Schematic illustrations of the two morphological evolution routes with different mechanisms. **c**–**j** TEM images of the self-assembled structures by dispersing the P*t*BA-*b*-PHATMA in 2-PrOH at 80 °C for 1 h and cooling down to r.t. naturally: (**c**) spheres, (**d**) worm-like structures, (**e**) tubular structures, (**f**) intermediate state of tubes dividing into vesicles, (**g**) vesicular structures, (**h**) intermediate structures, and (**i**) thin fibrils. **j** AFM image of the thin fibrils and the corresponding height profile. Scale bars are 1 μm, except for the image (**h**) which is 500 nm.

belt-like structures, depending on the doping level. Most importantly, when fully doped, the mesogens would be organized very quickly, and thick fibrils were formed immediately when the solution was cooled down. Surprisingly, the assembly mechanism was completely altered from intramolecular chain shuffling and rearrangement for the pristine BCP to nucleation-growth mode for the highly doped BCP (Fig. 1b lower). Consequently, the resultant fibrils could be subjected to self-seeding procedures to produce highly uniform fibrils with tunable lengths, which would further undergo end-to-end coupling to form supramicellar hierarchical structures.

## Results

The diblock copolymer poly(*tert*-butyl acrylate)-*block*-poly(6-(3,6,7,10,11-pentakis(hexyloxy)−2-oxytriphenylene) hexyl methacrylate) (P*t*BA$_{102}$-*b*-PHATMA$_{22}$, Fig. 1a) was synthesized *via* sequential atom transfer radical polymerization (ATRP)[18]. The discotic HAT mesogen-containing monomer 6-(3,6,7,10,11-pentakis(hexyloxy)−2-oxytriphenylene)hexyl methacrylate (HATMA) and its homopolymer were synthesized according to literature (Supplementary Figs. 1a, b, 3–14, Supplementary Table 1)[19]. The detailed molecular characterizations of the diblock copolymer were included in Supplementary Figs. 1c, 15–19 and Supplementary Table 2.

Although discotic LC polymers combine orientational molecular packing and fluidic nature and have been found attractive for thermal[20], optical[21], and electronic devices[22], its solution-state self-assembly behaviors have been rarely explored[23–26]. The hexagonal columnar packing of HAT mesogens has been well understood. When they were attached as pendant groups for polyacrylates, the organization patterns would be mostly retained[27]. Meanwhile, their LC properties could be significantly influenced positively or negatively[28–30]. In the current system, the LC properties were severely impaired, probably due to the introduction of an additional rigid methacrylate backbone. Very similar results were also obtained in the current study that the phase transition of the PHATMA block could not be detected *via* normal differential scanning calorimetry (DSC) characterization (Supplementary Fig. 20). The LC ordering effect was so weak that the orderedness could not be restored within a reasonable time range during the cooling process. To observe the LC to amorphous phase transition, we had to anneal the sample at 35 °C for 24 h before the DSC characterization and record the results from the first heating scan (Supplementary Fig. 21). However, this weak LC ordering effect makes it possible to set up a delicate balance between different interactions during the micellization process, triggering the continuous morphological transitions over time.

For the self-assembly experiment, the diblock copolymer was directly dispersed in 2-propanol (2-PrOH, concentration = 0.1 mg/mL) at 80 °C for 1 h and cooled down naturally to room temperature (r.t., 21 °C) within 3 h. Intricate multiple micellar morphological transitions were subsequently observed (Fig. 1b upper) *via* transmission electron microscope (TEM). Aliquots were taken at different stages and dried on TEM grids within seconds before observation (Fig. 1c–j). Initially, at 80 °C, all the polymer was dispersed (Supplementary Fig. 22). When the solution was cooled down to 75 °C, spherical aggregates appeared (Fig. 1c). At 60 °C, these spherical aggregates fused into thick worm-like aggregates with diameter of around 100 nm (Fig. 1d). After the solution reached r.t., these worm-like aggregates were further swollen into tubular structures with dramatically increased diameter (~500 nm, Fig. 1e). The hollow tubular structure was verified with grayscale analysis (Supplementary Fig. 23). Interestingly, after 2 h at r.t., these tubular structures started to divide into vesicles (Fig. 1f), of which the process would finish within another 10 h (Fig. 1g). The vesicular structures were also confirmed by the observation of wrinkles from their TEM, atomic force microscopic (AFM) images, and similarly grayscale analysis (Supplementary Figs. 23 and 24). Moreover, these vesicles started to spin out thin fibrils after 120 h of storage (Fig. 1h).

These jellyfish-like structures transformed into thin fibrils exclusively in 120 days (Fig. 1i), for which a width of 25.6 ± 4.6 nm, and a height of 2.0 ± 0.1 nm were measured from their TEM and AFM images (Fig. 1j), respectively.

It is not too surprising for BCPs to undergo one or two morphological transitions spontaneously during self-assembly[31,32], but it has been never reported that so many distinct morphologies could be shown in a single self-assembly process. A plausible driving force would be the subtle LC ordering effect from the HAT mesogens. The ordering of the LC phase was obviously slower than in the bulk[30,33], probably due to the plasticizing effect from the surrounding solvent molecules. So, it is possible that the LC ordering effect was more complicated in the morphological transition process due to the higher fluidity in the solution and interactions with solvent molecules. To better understand its role during the morphological transition, the fluorescence (FL) emission of HAT moieties was first monitored to look for a hint from their molecular packings (Fig. 2a). After the solution reached r.t., the peak intensity at 390 nm (excitation wavelength = 270 nm) gradually decreased over time until 120 h, due to the enhanced π-π stacking from the slowly organizing HAT mesogens[34]. Afterwards, with the gradual formation of fibrils, the peak shifted toward a longer wavelength, due to the even stronger π-π stacking[35], indicating the further boosted orderedness of the discotic mesogens.

The wide-angle X-ray scattering (WAXS) spectra of the pure vesicles and fibrils were also obtained to explore the molecular packing of the mesogens. Both spectra showed hexagonal columnar packings of the HAT mesogens (Fig. 2b). However, the (110) peaks were not as obvious as they were from homopolymers (Supplementary Fig. 14), due to the strong disturbance from the amorphous P*t*BA block[36]. Closer examinations of the spectra revealed a more obvious peak for the thin fibril sample at q = 1.85 Å$^{-1}$, corresponding to the (001) plane of the columnar phase, suggesting a higher LC orderedness for the thin fibrils. Moreover, the peaks at q = 0.17 Å$^{-1}$ and 0.34 Å$^{-1}$, from the lamella packing of the columnar mesophase, also appeared to be more obvious for the fibril sample. Meanwhile, with the morphological transformation from vesicles to fibrils, the lengths of the *a*-axis decreased from 21.33 to 20.9 Å, and the $d_{001}$ also decreased from 3.53 to 3.40 Å (Supplementary Figs. 26 and 27). All these results suggested a higher LC orderedness for these fibrils.

The grazing incident wide-angle X-ray scattering (GI-WAXS) results showed that the HAT disks were stacked perpendicular to the vesicular membrane (Supplementary Fig. 28). Meanwhile, the orientational LC packing of discotic mesogens within the core of these thin fibrils was verified *via* infrared-AFM (AFM-IR)[37]. Topographic image (Fig. 2c) of the thin fibrils was first obtained, and AFM-IR spectra (Fig. 2d) were acquired with the IR beam both parallel ($I_{parallel}$) and perpendicular ($I_{perpendicular}$) to the fibril at the marked black spot in Fig. 2c (Supplementary Note 1, Supplementary Figs. 29 and 30)[38]. At the wavenumber of 1435 cm$^{-1}$, which corresponds to the in-plane bending vibration of C=C bonds on the HAT disks, the dichroic ratio (DR = $I_{parallel}$/$I_{perpendicular}$) values reached 3.3 (Fig. 2d), demonstrating that the discotic mesogens were preferably packed in columns perpendicular to the length of fibrils[37]. Therefore, these fibrillar micelles appear more likely very thin belts.

Consequently, a detailed route for the morphological transition on the molecular level could be proposed. Initially, spherical aggregates were formed to lower the interfacial energy, due to the reduced solubility of the PHATMA block. The HAT moieties were amorphous in the micellar core. Subsequently, the discotic HAT disks started to slowly pack into ordered structures. At this stage, the orderedness of the disks increased over time, and the density of the micellar core should increase and its volume decrease spontaneously. For the micellization process of coil-coil BCPs[39], this change should prefer the formation of spherical micelles over worm-like micelles. However, our results showed exactly the opposite trend. This counterintuitive

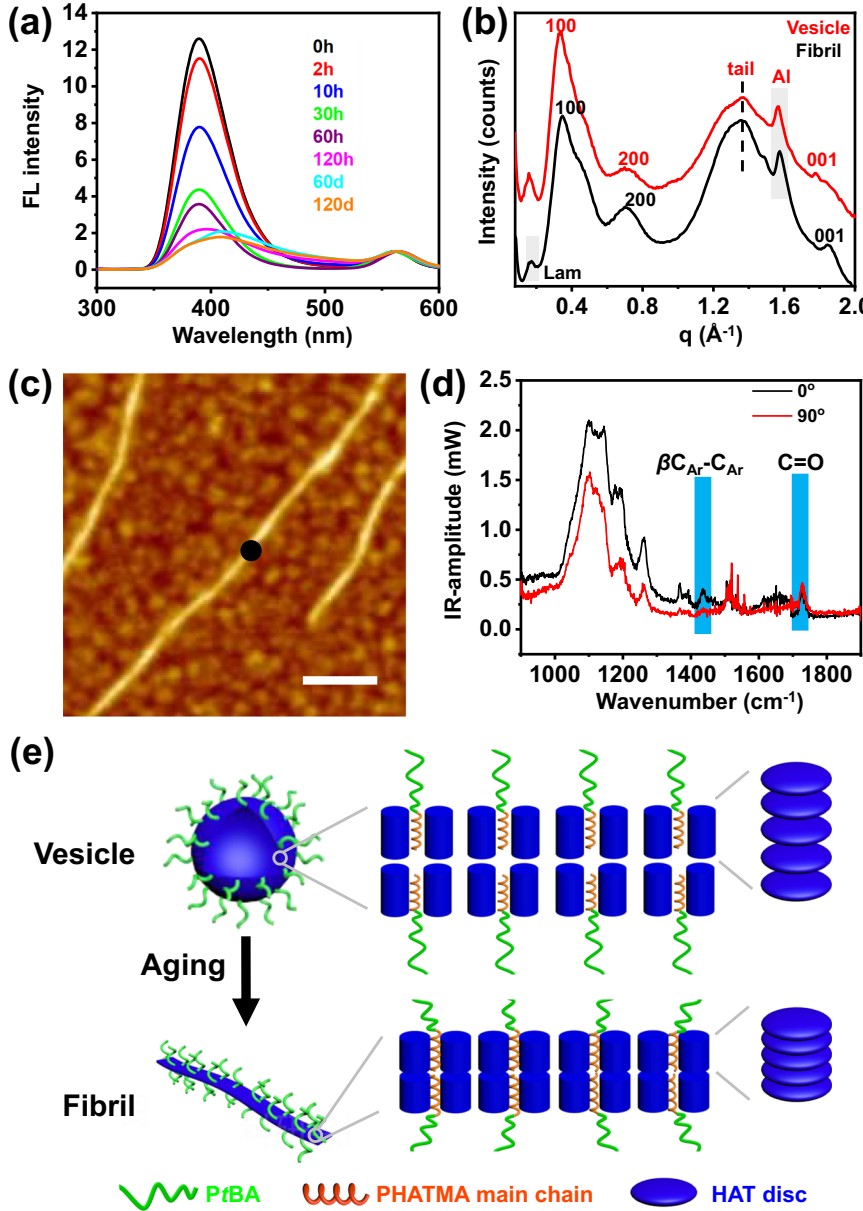

**Fig. 2 | Mesogenic ordering process. a** FL spectra of the 2-PrOH solution of P*t*BA-*b*-PHATMA assemblies at different times (rhodamine B as the internal standard at 565 nm). **b** Comparison of WAXS spectra of the vesicles, and thin fibrils. **c** Topographic image of thin fibrils obtained *via* AFM-IR. **d** Polarized AFM-IR spectra at the black spot indicated in **c. e** Schematic depictions of the molecular packings of the vesicle and fibrils. The scale bar is 100 nm.

observation clearly demonstrated that the subtle LC ordering effect from HAT moieties played a crucial role in the micellization process. During the next stage, the hexagonal columnar phase inside the vesicular membrane continued to pack even more closely. This would increase the crowdedness of the coronal chains, causing the further transformation from vesicles to fibrils (Fig. 2e)[32].

It is noteworthy that if the solution was quenched to r.t. within 1 min, the HAT mesogens were not able to organize into a well-defined columnar LC phase (Supplementary Fig. 31), resulting in random aggregates (Supplementary Fig. 32a). The poor flexibility of the PHATMA chains hindered the LC ordering process, and these random aggregates would not evolve into well-defined structures even after 6 months of storage (Supplementary Fig. 32b), demonstrating the vital role of the LC ordering effect in the morphological transition process.

Another especially interesting point for the LC ordering process is that the orderedness not only can be gradually built over time *via* intrinsic LC ordering effect, but also strongly enhanced very quickly by

mixing them with appropriate dopants[40,41]. Particularly, it has been reported that HAT mesogen can be doped with TNF (Supplementary Figs. 2 and 33) to form electron donor-acceptor complex (EDA), and significantly enhance the orderedness and thermal stability of the LC phase[42,43]. In the current case, when TNF was added to the tetrahydrofuran (THF) solution of PHATMA, the colorless solution turned black immediately within 3 seconds, due to the formation of EDA (Supplementary Fig. 34). The formation of EDA was confirmed with various spectrometric analyses as well (Supplementary Figs. 35–37). Therefore, for the following assembly experiment, the dopant was added to the THF solution of the diblock copolymer initially to allow them to fully complex with HAT mesogens, and the sample was dried under nitrogen flow before thermal dispersion into 2-PrOH.

To investigate the influence of doping level on the assembly morphology, we first designed a complicated experimental procedure to show that the dopants were complexed with the diblock copolymer quantitatively (Supplementary Note 2 and Supplementary Fig. 38).

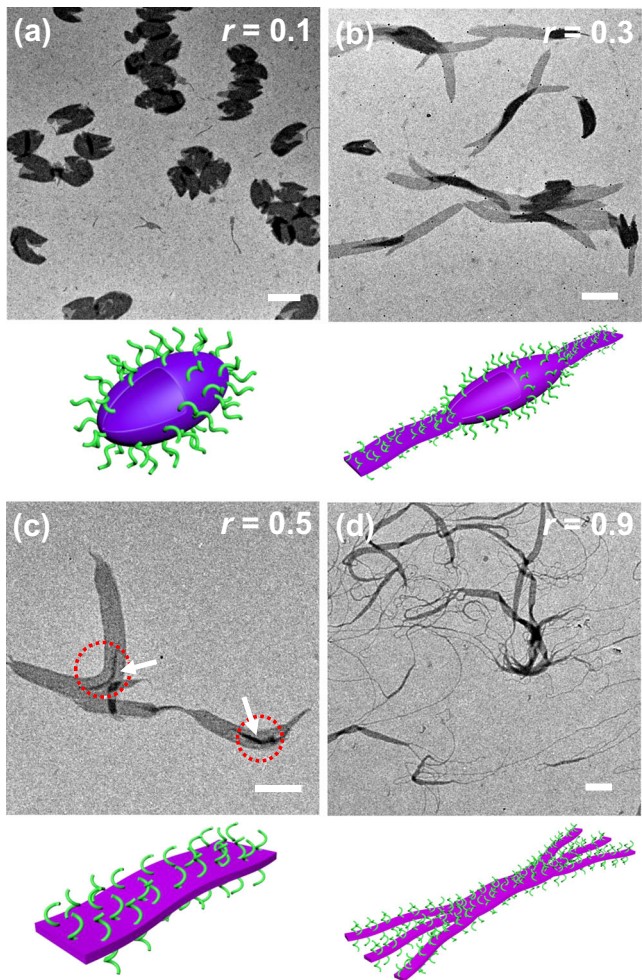

**Fig. 3 | Micelles obtained with doping. a–d** TEM images and the corresponding schematic illustrations of the self-assembled structures by dispersing the doped PtBA-*b*-PHATMA with various *r*-values in 2-PrOH at 80 °C for 1 h and cooling down to r.t. naturally. Scale bars are 1 μm.

Therefore, the doping level (*r* = mole of TNF/mole of HAT) could be tuned quantitatively. By increasing *r* from 0.01 to 1, not only did the isotropic to LC phase transition temperature increase (Supplementary Figs. 39 and 40), but also the *d*-spacing decreased accordingly (Supplementary Figs. 41 and 42), suggesting the enhancement LC orderedness with the increase of *r*.

The micellar morphologies were substantially influenced by doping (Fig. 3). With very small *r*-values (0.01–0.05, Supplementary Fig. 43), the spherical vesicles (Supplementary Fig. 44) would not completely transform into fibrils, and this jellyfish-like morphology was retained even after one-year storage at r.t. (Supplementary Fig. 45). Interestingly, when *r*-values increased over 0.05, elongated vesicles were obtained (Fig. 3a and Supplementary Fig. 46), which were formed *via* the deformation of spherical vesicles during cooling (Supplementary Figs. 47–49), and their aspect ratio increased with *r*-values to above 2 (Supplementary Fig. 50). Their morphological transition was also halted during the spinning of fibrils (Supplementary Fig. 51). With the increasing doping level, the rigidity of the LC PHATMA domain also increased, these vesicles had to elongate and release the bending penalty by forming topological defects at the vertices of the vesicles[44,45].

When the *r*-values reached 0.3, the membrane became too rigid that it could not be confined into closed vesicular structures, but instead flat bilayer structures were preferred to accommodate the high

bending penalty of the mesogens. As shown in Fig. 3b, and Supplementary Fig. 52, broken vesicles were observed, from the edge of which some belt-like structures stretched out. With further increasing of *r* values, the assemblies would transform toward belt-like structures, which were exclusively observed when *r* reached 0.5 (Fig. 3c and Supplementary Fig. 53). More interestingly, most of these belts appeared to be bent instead of straight as indicated by the red circles in Fig. 3c, Supplementary Figs. 53 and 64b, which has been rarely observed[46]. Additionally, some rod-like structures could be visualized within the belt-like structures, as indicated by the white arrows in Fig. 3c. With further increase of *r*, the belt-like assemblies transformed toward fibrils (Fig. 3d and Supplementary Fig. 54).

Surprisingly, when *r* = 1, exclusive fibrillar structures were obtained immediately after the solution was cooled to r.t. (Fig. 4). These fibrils appeared to be similar to those obtained from the undoped BCP, except that these fibrils were significantly thicker than the undoped ones, with a width of $45.1 \pm 7.0$ nm (Fig. 4a) from TEM images and a height of $4.0 \pm 0.3$ nm (Fig. 4b) from AFM images. Closer examinations revealed that the *d*-spacing of the doped fibrils (3.34 Å, Supplementary Fig. 55) was also smaller than that of the undoped ones (3.40 Å), suggesting a higher LC orderedness (Fig. 4e, Supplementary Fig. 56). With doping, the TNF molecules were inserted between HAT discotic mesogens to form an alternatively-layered structure, leading to thicker fibrils. Additionally, the higher LC orderedness inside these doped fibrils also led to an obviously higher rigidity, as indicated by their larger persistence length ($14048 \pm 332$ nm) than that of the undoped fibrils ($4405 \pm 149$ nm) (Supplementary Fig. 57, Supplementary Note 3)[47].

The enhancement in LC orderedness within the micellar core induced by TNF doping could be confirmed with the WAXS results (Supplementary Fig. 58), which agreed with those from TNF-doped PHATMA homopolymers (Supplementary Figs. 41, and 40). This unambiguously confirmed the addition of a dopant could markedly enhance the LC orderedness, which plausibly drove the formation of these unusual micellar morphologies (Fig. 1b lower). Very similar phenomena were reported in the previous study that LC orderedness could be greatly enhanced by doping the mesogens with small-molecule dopants, allowing the induction of mesophases even in previously non-liquid-crystalline polymers[48]. However, these studies only investigated their self-assembly behaviors in the bulk state, but no solution-state behavior has ever been explored.

It has been well established that the orientational alignment of discotic mesogens would be beneficial for electronic devices[49], and the formation of EDA could further enhance the conductivity[50]. As a proof of concept, the conductivity of the doped and undoped fibrils in the *z*-axis was measured with conductive AFM (c-AFM)[51]. As shown in Fig. 4c, the doped fibrils showed obviously enhanced conductivity compared to the undoped fibrils, confirming our hypothesis.

Moreover, it was found strangely that initially at high temperatures, instead of forming multiple morphological transitions, short rods were observed (Supplementary Fig. 59a). With slightly lower solution temperatures, the rods started to grow rapidly in length, resulting in long fibrils eventually (Supplementary Fig. 59b–h). The lengths of these fibrils were reasonably uniform and could be measured before they became too long after the solution temperature dropped below 50 °C (Supplementary Fig. 59). By calculating their number-average lengths ($L_n$), weight-average lengths ($L_w$), length polydispersity (PDI = $L_w/L_n$), and plotting the $L_n$ and PDI with temperature, a typical exponential growth curve was observed (Fig. 4d, Supplementary Table 3). Meanwhile, their PDI remained quite low during the growth process. This morphological evolution process closely resembled the in-situ nucleation-growth process[52–54], completely different from the undoped case.

Subsequently, based on such nucleation-growth assembly mechanism, these doped fibrils were subjected to self-seeding

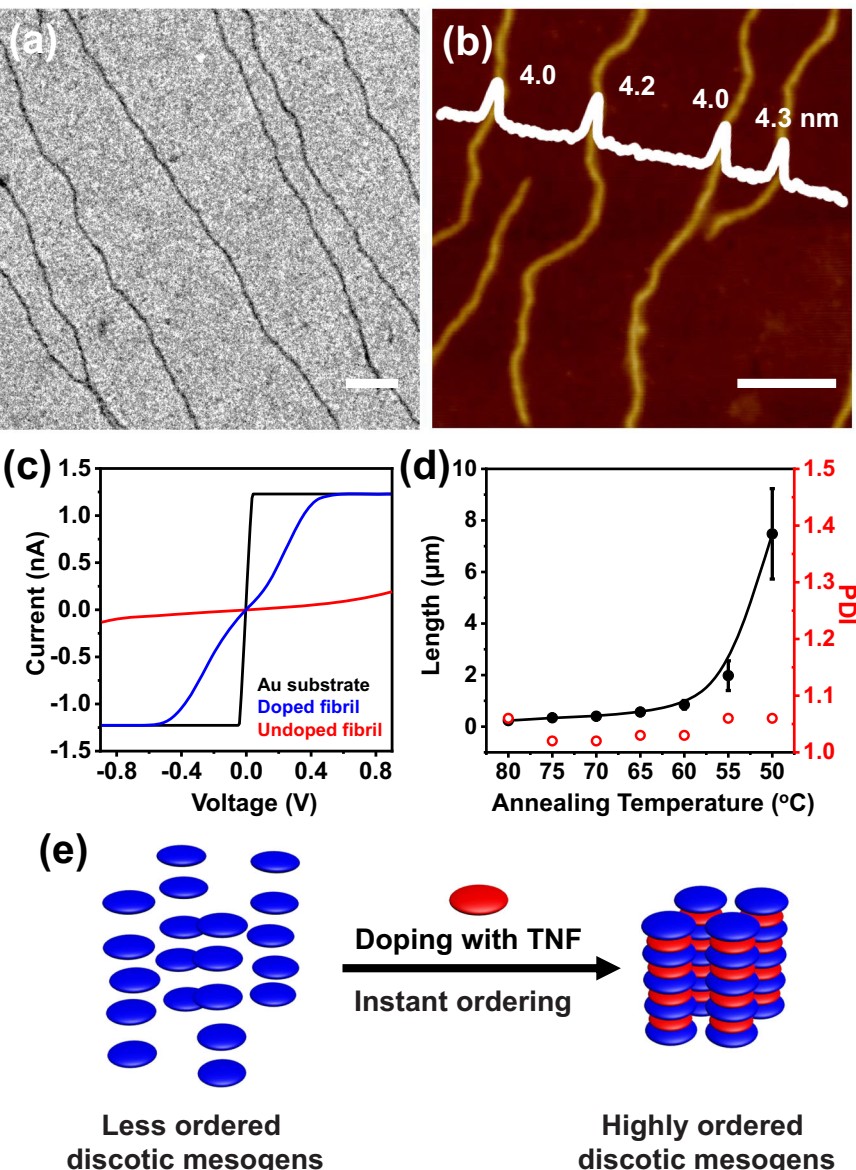

**Fig. 4 | Doped fibrils.** TEM (**a**) and AFM topographic (**b**) images of the thick fibrils formed by dispersing the pre-doped TNF/P*t*BA-*b*-PHATMA (*r* = 1) in 2-PrOH at 80 °C for 1 h and cooling down to r.t. naturally. **c** The comparison of the conductivity from doped and undoped fibrils measured in the *z*-axis with c-AFM. **d** Variation of the fibril lengths versus solution temperatures. The detailed length information of all

the samples is summarized in Supplementary Table 3. Error bars represent mean ± standard deviation, *n* ≥ 200. **e** Schematic depictions of the doping of HAT moieties with TNF molecules and the corresponding packing of discotic mesogens. Scale bars are 500 nm.

procedures to produce highly uniform fibrils. These pristine fully doped fibrils were first fragmented *via* ultrasonication at 0 °C to yield short rods (seeds, Supplementary Fig. 60). The $L_n$ was determined to be 121 nm from their TEM images. Seed solutions (total concentration = 0.005 mg/mL) were thermally annealed for 1 h at desired temperatures from 30 °C to 72 °C, and the annealed solutions then were cooled naturally to r.t. The rods with lower LC orderedness dissolved upon heating, while those with higher orderedness survived and functioned as seeds for the dissolved chains to grow epitaxially during the cooling process[53]. With an increasing annealing temperature from 30 °C to 72 °C, the fibrils with narrow length distributions (PDI ≤ 1.04) were obtained and their lengths increased accordingly (Supplementary Fig. 61; Supplementary Table 4). Typical TEM images for the samples annealed at 65 °C ($L_n$ = 270 nm, PDI = 1.03) and 72 °C ($L_n$ = 611 nm, PDI = 1.02) were included in Fig. 5a, b. The length information for samples at different annealing temperatures was

summarized in Fig. 5c. The fraction of surviving seeds at each annealing temperature was calculated as well (Supplementary Note 4). As shown in Fig. 5c, in the range of 60–72 °C, the fraction of surviving seeds decreased exponentially with increasing temperature, a key characteristic of a self-seeding process (Fig. 5e, Supplementary Note 5, Supplementary Fig. 62)[55]. Interestingly, after being aged for a long period, a large portion of these fibrils underwent end-to-end coupling to form segmented supramolecular micelles (Fig. 5d, e), similar to our findings in other LC BCP systems[56]. In sharp contrast, for the undoped fibrils, heating only led to their dissolution and restart of the morphological transition (Supplementary Fig. 63).

Consequently, a plausible explanation for the formation of bent belt-like structures could also be proposed. Similar to the case of thick fibrils, short rods were observed initially at 80 °C (Supplementary Fig. 64a), due to the existence of highly doped diblock copolymers and the formation of a highly ordered LC phase. These short rods

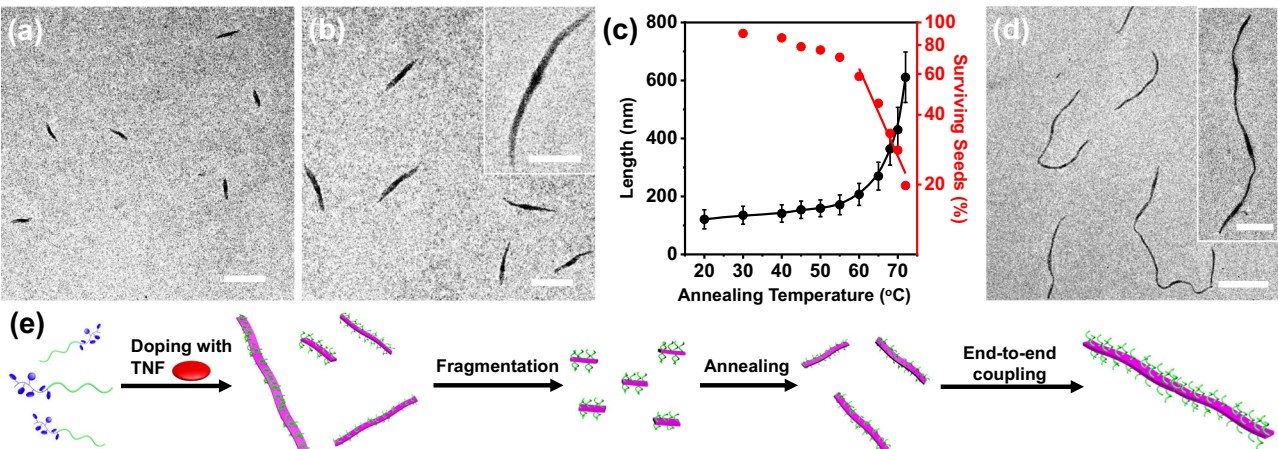

**Fig. 5 | Self-seeding of doped fibrils.** TEM images of the fibrils after self-seeding experiments with annealing temperatures at (**a**) 65 °C, and (**b**) 72 °C. (Scale bars are 500 nm in the images and 200 nm in the inset.) **c** The plot of $L_n$ and the semilogarithmic plot of the fraction of surviving seeds in solution *versus* annealing temperatures. The red line represents the best linear fit for the data points from 60 °C to 72 °C. The detailed length information of all the samples is summarized in Supplementary Table 4. Error bars represent mean ± standard deviation, $n \geq 200$. **d** TEM image of the end-to-end coupled fibrils. (Scale bars are 1 μm in the image and 500 nm in the inset.) **e** Schematic illustration of the self-seeding and the subsequent end-to-end coupling process.

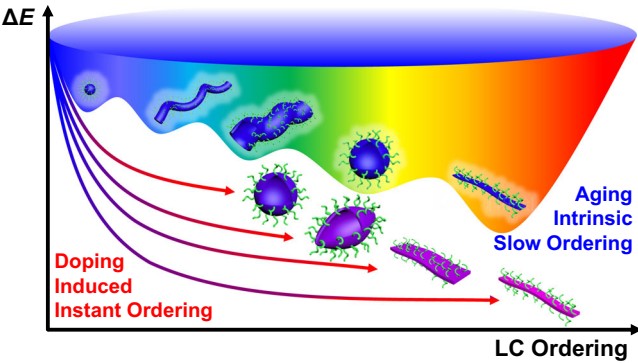

**Fig. 6 | Convoluted morphological transitions.** Energetic landscape of the morphological evolution *via* aging (intrinsic slow ordering) or doping (induced quick ordering) routes.

functioned as seeds to trigger the growth of the less-doped BCP chains during the cooling process (Supplementary Fig. 64b). The unsaturated doping caused the nonuniform distribution of rigidity and stress inside the micellar core, eventually leading to the formation of bent belts. Similarly, self-seeding procedures could be applied to belt-like structures (Supplementary Fig. 65) to obtain uniform belt-like structures (Supplementary Fig. 65), strongly demonstrating the nucleation-growth mode again.

However, it was noteworthy that if TNF was added to the micellar solution, the morphological transitions would not be influenced (Supplementary Fig. 66). Even with the excessive TNF ($r = 10$), the vesicular micelles still slowly evolved into jellyfish-like structures, and eventually thin fibrils within the same range of time as those without doping. This demonstrated that TNF molecules were not able to enter the mesogenic micellar core if they were not complexed with HAT prior to the assembly process. Therefore, such morphological transition should occur *via* the intramolecular chain shuffling and rearrangement, without involving individual BCP chains and dissociation/reassociation of the assemblies[57]. On the contrary, with doping, the formation of EDA complexes completely changed the assembly mechanism to nucleation-growth mode. LC orderedness of the diblock copolymer could be very quickly built by doping the mesogens with small-molecule dopants, and the morphological transitions were dramatically accelerated. The adoption of small-molecule dopant TNF plausibly facilitated the system to reach a thermodynamic equilibrium

state. Therefore, the assembly route was transformed from kinetic-controlled for the pristine diblock copolymer into thermodynamic-favored nucleation-growth mode at or close to the equilibrium state. It could be envisioned that if TNF was introduced as a side group in side-chain polymer[58], it may exhibit quite different kinetic behavior. The morphological transition was no longer a result of the balance between different energy items but was dictated by the LC ordering effect. Such doping-induced alteration of the assembly mechanism was not seen before.

Lastly, some comprehensive understandings can be concluded on the solution self-assembly behaviors of this discotic mesogen-containing diblock copolymer. As shown in Fig. 6, the energetic landscape and pathway of this assembly system are highly convoluted. The LC orderedness can be achieved in two different approaches. Due to the subtle intrinsic LC ordering effect from this PHATMA block, the LC phase can be slowly built over time. Consequently, the micellar morphology results from the delicate balance between different energetic items, such as the configurational entropy change of the core block after compression in the micellar core, the interface energy reduction *via* micellization, and the energy lost by stretching of the corona block[32], leading to slow morphological transitions to reveal multiple metastable micellar morphologies. Meanwhile, the LC ordering effect can be significantly enhanced upon doping to strongly influence or even dominate the morphological transition. With low doping levels, dopants will cause local orderedness variations and interfere with the morphological transitions, yielding exotic morphologies. With high doping levels, due to the uneven distribution of dopants, the highly-doped diblock copolymer can first aggregate to form mesogenic domains, which subsequently function as seeds to trigger the growth of the less-doped polymer chains. Finally, if the mesogens are fully doped, thick fibrils will be formed *via* a typical in-situ nucleation-growth mode, driven by the enhanced LC ordering effect. Such a doping-induced route to enhance the organization ability of assembly motifs presents a versatile, and efficient strategy to achieve precisely controllable assembly from systems with weak ordering effects. An external component is added to enhance the organization ability of the assembly motifs, of which the underlying rationale is quite similar to the controllable assembly of supramolecular motifs based on guest/host pairs[52,59]. We envisage that it can be widely applied to many other assembly systems and some interesting features such as chiral assemblies can be readily realized *via* chiral doping[60].

## Discussions

Herein, we reported the convoluted micellar morphological transitions driven by the subtle LC ordering effect from a discotic mesogen-containing BCP. Interestingly, if the assembly process was initiated by directly dispersing the polymer in solvent *via* heating, the diblock copolymer would undergo unusual multiple morphological transitions over a very long period, driven by the delicate balance of energetic items and the intrinsic slowly building LC orderedness. Surprisingly, if the discotic mesogens were doped to form electron donor-accept complexes, their LC orderedness would be built very quickly, which could significantly influence the assembly process to yield various exotic micellar morphologies. Most importantly, with high doping levels, the self-assembly behaviors of the diblock copolymer were completely altered from intramolecular chain shuffling and rearrangement to nucleation-growth mechanism, allowing the precise control of fibril length *via* self-seeding procedures. A doping-induced alteration of assembly behavior was discovered, and this surprising finding revealed an unknown approach of doping-triggered controllable assembly. We believe this study not only expands the repertoire of solution-state self-assembly of BCPs, providing mechanistic insights into the crucial role of LC ordering in complicated micellization processes but also suggests a unique approach for generating nanostructures in a highly controllable manner.

## Methods

### Materials

Catechol (99%), 1-bromohexane (99%), potassium carbonate($K_2CO_3$, 99%), potassium iodide (KI, 99%), light petroleum (PE, 99%), methanol (MeOH, 99%), ethanol (EtOH, 99%), dichloromethane ($CH_2Cl_2$, 99%), tetrahydrofuran (THF, 99%), iron (III) trichloride (99%), boron tribromide ($BBr_3$, 99%), sodium sulfate ($Na_2SO_4$, 99%), 6-bromo-1-hexanol (99%), acetonitrile ($CH_3CN$, 99%), calcium hydride ($CaH_2$, 99%) 1,4-dioxane (99%), triethylamine (TEA, 99%), methacryoyl chloride (99%), cuprous bromide (99%, CuBr), *N,N,N',N'',N''*-pentamethyldiethylenetriamine (PMDETA, 99%), *tert*-butyl acrylate (*t*BA, 99%), the initiator 2-hydroxyethyl 2-bromoisobutyrate (HEBiB, 99%) and ethyl acetate (99%) were purchased from Aldrich and were used as received unless otherwise stated. Nitric acid ($HNO_3$, 65–68%), sulfuric acid ($H_2SO_4$, 98%), acetone (99%), and toluene (99%) were purchased from Beijing Chemical Works. Silica gel 60 (200–300 mesh ASTM, 99%), neutral aluminum oxide ($Al_2O_3$, 200–300 mesh, activated, 99%), and silica gel 60 glass thin-layer chromatography were used for the purification and identification of the reaction, respectively. The *t*BA monomer, $CH_2Cl_2$, THF, and toluene were distilled over $CaH_2$ before use. All other solvents were used as received without further purification. For the atom transfer radical polymerization (ATRP), CuBr was purified with acetic acid before use. All of the self-assembly experiments were performed in HPLC-grade solvents that were acquired from Fisher.

### Synthesis of P*t*BA-*b*-PHATMA

The diblock copolymers were synthesized *via* sequential polymerization of the two monomers, *t*BA and HATMA *via* the ATRP method (Supplementary Fig. 1c). CuBr (36.0 mg, 0.25 mmol), PMDETA (52.5 μL, 0.25 mmol), *t*BA (5.44 mL, 37.5 mmol), HEBiB (38.25 μL, 0.25 mmol) and 3 mL toluene were introduced into a Schlenk tube and degassed with three freeze-pump-thaw cycles. Subsequently, the polymerization solution was heated at 60 °C for 6 h under a nitrogen atmosphere with vigorous stirring. Cupreous salt was removed by filtering the reaction solution through $Al_2O_3$ columns, and the polymer was further purified *via* repeated precipitations from THF solution into a mixture of water and methanol (7/3, volume ratio), and dried under reduced pressure. A white solid of P*t*BA was obtained (2.19 g, yield 45%).

To polymerize the second block, P*t*BA macroinitiator (260 mg, 0.02 mmol), CuBr (5.8 mg, 0.04 mmol), PMDETA (8.4 μL, 0.04 mmol),

6-[3,6,7,10,11-pentakis(hexyloxy)−2-oxytriphenylene]hexyl methacrylate (182.6 mg, 0.2 mmol) and 2 mL 1,4-dioxane were introduced into a Schlenk tube and degassed with three freeze-pump-thaw cycles. Subsequently, the polymerization solution was heated at 80 °C for 6 h under a nitrogen atmosphere with vigorous stirring. The reaction mixture was purified *via* column chromatography with PE/ethyl acetate (silica gel, 10/1, volume ratio) as the eluent first to remove the residual trace monomer and then the polymer was eluted with ethyl acetate, the concentrated crude product was further purified *via* repeated precipitations from their THF solution into MeOH, and dried under reduced pressure. A white solid was obtained as the final product (160 mg, yield 36%).

**Polymer characterization.** Molecular weight and polydispersity indices of polymers were obtained with a Viscotex gel permeation chromatography (GPC) max chromatograph equipped with styrene/divinylbenzene columns with pore sizes of 500 Å and 100,000 Å, VE 3580 refractometer. THF (Fisher) was used as the eluent, with a flow rate of 1.0 mL/min. Samples were dissolved in the eluent (10 mg/mL) and filtered (Acrodisc, PTFE membrane, 0.45 μm) before analysis. The calibration of the refractive index detector was carried out using polystyrene standards (Viscotek). To determine the molecular weight of the BCPs, aliquots of the first block were taken and the absolute molecular weight of the first block was determined by matrix-assisted laser desorption/ionization-time of flight mass spectrometry (MALDI-TOF). The absolute polymerization degrees of the two blocks were then determined by combining the molecular weight $M_n$ of the first block from MALDI-TOF spectrometry measurements with the block ratio of the diblock copolymer obtained by integration of the $^1H$ NMR spectrum.

### Transmission electron microscopy (TEM)

TEM samples were prepared *via* drop-casting approximately 5 μL of the micellar solution onto a carbon-coated copper grid (Beijing Zhongjingkeyi Technology Co., Ltd., mesh 230). To swiftly eliminate excess solvent and prevent any further changes in morphology, grids were pre-positioned on filter paper for 1 second. Bright-field TEM micrographs were captured using a Hitachi H-7650B microscope operating at 80 kV. No staining was administered to TEM samples unless specified otherwise. Image analysis was conducted using the ImageJ software package developed by the US National Institute of Health. For each sample, ca. 100 micelles in several images were traced by hand to obtain the length information. The number average micelle length ($L_n$) and weight average micelle length ($L_w$) were calculated using Eqs. (1) and (2), from measurements of the contour lengths ($L_i$) of individual micelles, where $N_i$ is the number of micelles of length $L_i$, and n is the number of micelles examined in each sample.

$$L_n = \frac{\sum_{i=1}^{n} N_i L_i}{\sum_{i=1}^{n} N_i} \tag{1}$$

$$L_w = \frac{\sum_{i=1}^{n} N_i L_i^2}{\sum_{i=1}^{n} N_i L_i} \tag{2}$$

The distribution of micelle lengths is characterized by both $L_w/L_n$ and the standard deviation of the length σ.

### Atomic force microscopy (AFM)

AFM images were captured using a Dimension FastScan instrument from Bruker. The experiments involved either direct imaging on a carbon-coated copper grid utilized for TEM analysis or preparation *via* drop-casting the solution onto a mica substrate. Imaging was conducted in tapping mode under an ambient environmental condition.

## Conductive Atomic force microscopy (c-AFM)

c-AFM images were recorded by using Dimension IconIR (Bruker). The electroactive nature of the doped and undoped fibrils was demonstrated using tunneling experiments. Fibril dispersions (0.1 mg/mL in 2-PrOH) were deposited *via* spin-coating on silicon substrates coated with an Au layer (30 nm).

## Atomic force microscopy-based infrared spectroscopy (AFM-IR)

AFM-IR was recorded by using Dimension IconIR (Bruker). AFM-IR spectra of parallel and perpendicular polarizations were collected at the same location on the micelles by turning the polarizer to 0 or 90°, respectively. Fibril dispersions (0.1 mg/mL in 2-PrOH) were deposited *via* spin-coating on silicon substrates coated with an Au layer (30 nm).

## Wide-angle X-ray scattering (WAXS)

WAXS of the samples were recorded by employing the Ganesha system (SAXSLAB, U.S) equipped with a multilayer focused Cu Kα radiation as the X-ray source (Genix3D Cu ULD) and a semiconductor detector (Pilatus 300 K, DECTRIS, Swiss). A Linkam THMS600 hot stage was utilized to study the structure evolution as a function of temperature. The heating and cooling rates in the experiments were 10 °C/min.

## Data availability

The datasets generated during and/or analyzed during the current study are available from the corresponding author on request.

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

## Acknowledgements
X.Y.L. is grateful for the financial support from the National Natural Science Foundation of China (Grant numbers 51973019 and 22175024). B.X.J. is grateful for the financial support from the China Postdoctoral Science Foundation (No. 2021TQ0033) and the National Natural Science Foundation of China (Grant number 52303266). The grazing-incident wide-angle X-ray scattering data were obtained at the beamline BL16B1 of Shanghai Synchrotron Radiation Facility (SSRF).

## Author contributions
The project was conceived and designed by H.Y., B.J. Y.L. and X.L. The experimental results were obtained by H.Y., B.J. and Y.L., and H.Y. was responsible for materials synthesis. S.C. performed in AFM experiments. The manuscript was written by H.Y., B.J. and X.L. with useful input from the other co-authors.

## Competing interests
The authors declare no competing interests.
