## [Peer Review File · Nature Communications]

Convoluting Micellar Morphological Transitions Driven by Tailorable Mesogenic Ordering Effect from Discotic Mesogen-Containing Block CopolymerREVIEWER COMMENTS

Reviewer #1 (Remarks to the Author):

The authors report on the complex morphological evolution of a diblock copolymer in which one of the segments is a side-chain discotic LC polymer. The results reported in this work are of broad interest, as they underscore the complex interplay of ordering across different length scales. Of particular note is the observed acceleration and changes that result from doping the samples with trinitrofluorenone (TNF). The authors have carried out an extensive and thorough examination of these systems, so much so that my major critique is that the present work goes beyond what I would regard as a communication. The authors, in an apparent effort at brevity, have in many respects not done justice to their studies and analyses. Indeed, many rather complex data sets are described so sparingly that it is difficult to follow the arguments. In particular, the proposed ordering of the discs within the fibrils was not clearly described. What, in particular, is meant by belt-like structures? How do they envision the packing of the columns-- are these effectively narrow sheets? Is there curvature? What are the dimensions and how do these relate to the molecular dimensions and the packing typically seen in HAT mesophases? What do the authors mean when they talk about lamellar ordering of columnar mesogens (note: columnar refers to a kind of mesophase, not a type of mesogen)?

One other point that I believe should be addressed. The authors cite the work of Ringsdorf and acknowledge his work on DA complexes of TNF and HAT from several decades ago. Nonetheless, I did not feel that the authors have properly detailed the extent to which observations in present work were anticipated by Ringsdorf's extensive studies. In particular, many of the effects of TNF/HAT complexation (shorter distances, higher ordering) were previously noted in non-polymeric systems. This is not to diminish the novelty of some of the key observations in the present work, as Ringsdorf's work was not concerned with solution properties nor block copolymers.

Finally, I found the repeated use of the term "instantly" to describe the effects of TNF on morphology inappropriate; a more scientific approach to describing timescales is in order.

Reviewer #2 (Remarks to the Author):

In this manuscript the study of the evolution of convoluted micellar morphologies and their transitions in triphenylene discotic containing block copolymers have been reported. Multiple morphological transitions have been observed in these mesogenic block copolymer systems. Moreover, the authors have introduced charge transfer complex forming dopants in the system and observed that the ordering and morphological transitions can be accelerated and better ordering of the mesogens can be obtained by the doping method. With high doping concentrations, the mode of self-assembly of the block copolymers in solution changes and it is possible to obtain uniform fibrils. Interesting insights have been gained on the solution state self-assembly of the discotic mesogen containing diblock copolymer. The experimental observations of different micellar transitions have been fairly characterized and rationalized in the manuscript. However, the following should be clarified and addressed before its

publication.

Does the mesogen functionalized block copolymer form any lyotropic liquid crystal phase at higher concentrations than used in the study ?

The authors mention that “work-like aggregates were further swollen into tubular structures”. How did the authors confirm the tubular nature of the aggregates ? Experimental data should be provided for this claim i.e., TEM or SEM images.

Similarly, they state the tubular structures developed into vesicles. I do not see evidence for the vesicle structures. How did they differentiate between micellar structure and vesicular morphologies ?? Electron density map calculations might be of help.

The XRD data in figure 2b seems to indicate a columnar lamellar ordering but the authors claim the occurrence of hexagonal columnar ordering. The exact nature of the mesophase should be resolved probably using fast Fourier transform (FFT) of the TEM images which could reveal the underlying ordering in the self-assembled structure.

Reviewer #3 (Remarks to the Author):

Solution self-assembly of synthetic well-defined copolymers with hierarchical structures and various rich morphologies even showing some interesting functionalities have attracted tremendous attention for quite a few years, thanks to its important significance for better understanding or mimicking the complicated and delicate natural assembly behaviors with complex biological functions. This work by Prof. Xiaoyu Li group presents a systematic investigation of solution-state self-assemblies of diblock copolymer with triphenylene (TP) discotic mesogens. It is very interesting to find that the diblock copolymer of poly(tert-butyl acrylate)-block-poly(TP methacrylate) spontaneously underwent multiple morphological transitions with the intrinsic LC ordering. What is even more impressive is that it seems the LC order can be instantly built through doping the diblock copolymer with a small-molecule dopant of 2,4,7-trinitrofluorenone (TNF). A dramatically accelerated orderly organization process has been demonstrated with the assembly mechanism markedly changed from intramicellar chain shuffling and rearrangement for the pristine diblock copolymer to nucleation-growth mode for the highly doped complexes, which account for producing highly uniform fibrils with tunable lengths via self-seeding procedures. Overall, the manuscript presents a well conducted nice work, after some revision or alterations as pointed below, I am pleased to recommend its acceptance to publish in the prestigious top journal of Nature Communications.

Some more comments:

(1) Though the solution self-assembly of synthetic copolymers seems to perform a variant kinetics as compared to the bulk assembly of the same comparable copolymers referring to *Macromolecules* 2015, 48, 6768-6780, for TP-based homopolymers and *J Polym Sci, Part A: Polym Chem* 2017, 55, 2544-2553 for TP-based diblock copolymers.

(2) The assembly mechanism change into nucleation-growth mode for the highly doped complexes may be not so surprising, in my opinion, it reflects a transformation from a kinetic controlled process for the pristine TP-based diblock copolymer into nucleation-growth mode of thermodynamic at or close to the equilibrium state, which may be ascribed to the adoption of small-molecule dopant TNF facilitating the

ease in reaching thermodynamic equilibrium state. Otherwise, if TNF was introduced as side group in side-chain polymer as did by Percec et al. (Nature, 2002, 419, 384-387), it may exhibit quite different kinetic behavior.

The situation mentioned above in (1) and (2) is suggested to be incorporated into the discussions for kinetics versus thermodynamic processes, and compared to the bulk assembly.

(3) "When they were attached as pendant groups for polyacrylates, the organization patterns were mostly retained,^{27, 28} but their LC properties were severely impaired, due to the limited mobility from the polyacrylate backbone.²⁹" This sentence is an untimely and incorrect statement from ref.²⁷ (year 2000), ref.²⁹ (year 1989) with some misunderstanding, not in line with the updated cognition for discotic side-chain liquid crystalline polymers as renewed in ref. 28 and *Macromolecules* 2015, 48, 6768-6780 revealing well-organized columnar superlattices via positive coupling between polymer backbone and discotic side groups. Actually, TP-based discotic columnar liquid-crystalline polymer semiconducting materials via rational macromolecular engineering showed high charge-carrier mobility, which was attributed to significantly enhanced orientational correlation along polymer backbone, together with dynamic disorder inhibition and easy columnar alignment, though the restricted rotational and translational movements in side-chain discotic liquid crystalline polymers (*Polym. Chem.* 2017, 8, 3286-3293).

(4) A closely related reference (*Polym. Chem.* 2017, 8, 3457-3463) should be additionally cited, which demonstrated self-assembled helical columnar superstructures with selective homochirality from TP-based side-chain discotic liquid crystalline polymer doped with chiral acceptor molecules of ester derivatives of TNF to construct electron donor-acceptor (EDA) complexes through CT interactions.

(5) "for which a diameter of 25.6 ± 4.6 nm, and a height of 2.0 ± 0.1 nm were measured..." "with a diameter of 45.1 ± 7.0 nm (Fig. 4a)..." Here, for belt-like morphologies, width may be more suitable than diameter?

(6) "At this stage, the orderedness of the disks inclined over time,..." "With an inclining annealing temperature from 30 oC to 72 oC, the fibrils..." Here, "inclined", "inclining", seems to be changed into increased, increasing?

(7) In the Supporting Information file, in the ¹H NMR spectra of Figure S1-S6, S28, and of Figure S9, S13, S30, the labelled CDCl₃ or CD₂Cl₂ is improper, those peaks are from the residual non-deuterated solvents, deuterated solvents have no ¹H NMR signal! In Figure S27, the caption "(b) then stored storing for 6 months at r. t." change into "(b) then stored for 6 months at r. t."

Minor Revision.

Re: Manuscript number: NCOMMS-23-40393A-Z

Title: "Convoluting Micellar Morphological Transitions Driven by Tailorable Mesogenic Ordering Effect from Discotic Mesogen-Containing Block Copolymer"

Changes to the manuscript and supplementary information are highlighted in yellow.

Reviewer 1:

The reviewer makes six suggestions:

Comment 1: *The authors have carried out an extensive and thorough examination of these systems, so much so that my major critique is that the present work goes beyond what I would regard as a communication. The authors, in an apparent effort at brevity, have in many respects not done justice to their studies and analyses.*

Response: We thank the reviewer for pointing this out. Indeed, we have carried out a comprehensive study of the current system. However, I believe it fits the manuscript length requirement from the journal. As it states on the website of Nature Communications, "**The main text (not including abstract, Methods, References and figure legends) is limited to 5,000 words.**" and "**Articles may have up to 10 display items (figures and/or tables).**". The current version of the manuscript contains only 3700 words and 6 figures.

Comment 2: *Indeed, many rather complex data sets are described so sparingly that it is difficult to follow the arguments. In particular, the proposed ordering of the discs within the fibrils was not clearly described.*

Response: This is an important issue and we thank the reviewer for pointing this out. Actually, we have already carried out many experiments and included many discussions in the original manuscript to provide a comprehensive understanding of the molecular packings of the discotic mesogens.

The wide-angle X-ray scattering (WAXS) results of the pure fibrils were obtained and showed hexagonal columnar packings of the HAT mesogens, as shown in Fig. 2b in the original manuscript. The peaks at $q = 0.17 \text{ \AA}^{-1}$ and 0.34 \AA^{-1} were from the lamella packing of the columnar mesophase of the fibril sample. Closer examinations of the spectra revealed a more obvious peak for the thin fibril sample at $q = 1.85 \text{ \AA}^{-1}$ (3.34 \AA) which the d-spacing of the fibrils was π - π stacking distance, corresponding to the (001) plane of the columnar phase (Fig. 2b). It was discussed in the original manuscript as follows:

"However, the (110) peaks were not as obvious as they were from homopolymers (Fig. S12), due to the strong disturbance from the amorphous PtBA block. Closer examinations of the spectra revealed a more obvious peak for the thin fibril sample at $q = 1.85 \text{ \AA}^{-1}$, corresponding to the (001) plane of the columnar phase, suggesting a higher LC orderedness for the thin fibrils. Moreover, the peaks at $q = 0.17 \text{ \AA}^{-1}$ and 0.34 \AA^{-1} , from the lamella packing of the columnar mesophase, also appeared to be more obvious for the fibril sample. Meanwhile, with the morphological transformation from vesicles to fibrils, the lengths of the a-axis decreased from

21.33 to 20.9 Å, and the d_{001} also decreased from 3.53 to 3.40 Å (Fig. S24, and 25). All these results suggested a higher LC orderedness for these fibrils.”

In addition, the grazing incident wide-angle X-ray scattering (GI-WAXS) results showed that the HAT disks were stacked perpendicular to the vesicular membrane, as shown in Fig. S26 in the original supplementary information. It was discussed in the original manuscript as follows:

“The grazing incident wide-angle X-ray scattering (GI-WAXS) results showed that the HAT disks were stacked perpendicular to the vesicular membrane (Fig. S26).”

The height of the thin fibrils was measured from their transmission electron microscope (TEM) and atomic force microscopic (AFM) images (Fig. 1i and Fig. 1j), for which a diameter of 25.6 ± 4.6 nm, and a height of 2.0 ± 0.1 nm respectively. Combining the experimental results from TEM and AFM, we proposed that the height of the thin fibrils was the height of two discrete columnar stacks (DCS) containing about π - π stacking distance of ten discs. It was discussed in the original manuscript as follows:

“These jellyfish-like structures transformed into thin fibrils exclusively in 120 days (Fig. 1i), for which a width of 25.6 ± 4.6 nm, and a height of 2.0 ± 0.1 nm were measured from their TEM and atomic force microscopic (AFM) images (Fig. 1j), respectively.”

In addition, the orientational LC packing of discotic mesogens within the core of these thin fibrils was verified via infrared-AFM (AFM-IR). Topographic image (Fig. 2c) of the thin fibrils were first obtained, and AFM-IR spectra (Fig. 2d) were acquired with the IR beam both parallel (I_{parallel}) and perpendicular ($I_{\text{perpendicular}}$) to the fibril at the marked black spot in Fig. 2c (Supplementary Note 1 and Figs. S27, and 28). AFM-IR spectra demonstrated that the discotic mesogens were preferably packed in columns perpendicular to the length of fibrils. As shown in the schematic illustration in Fig. S25 in the original supplementary information, molecular packing of the discotic mesogens was described within the fibrils. It was discussed in the original manuscript as follows:

“Meanwhile, the orientational LC packing of discotic mesogens within the core of these thin fibrils was verified via infrared-AFM (AFM-IR). Topographic image (Fig. 2c) of the thin fibrils were first obtained, and AFM-IR spectra (Fig. 2d) were acquired with the IR beam both parallel (I_{parallel}) and perpendicular ($I_{\text{perpendicular}}$) to the fibril at the marked black spot in Fig. 2c (Supplementary Note 1 and Fig. S27, and 28). At the wavenumber of 1435 cm^{-1} , which corresponds to the in-plane bending vibration of C=C bonds on the HAT disks, the dichroic ratio ($DR = I_{\text{parallel}}/I_{\text{perpendicular}}$) values reached 3.3 (Fig. 2d), demonstrating that the discotic mesogens were preferably packed in columns perpendicular to the length of fibrils. Therefore, these fibrillar micelles appear more likely very thin belts.”

Comment 3: *What, in particular, is meant by belt-like structures? How do they envision the packing of the columns-- are these effectively narrow sheets? Is there curvature? What are the dimensions and how do these relate to the molecular dimensions and the packing typically seen in HAT mesophases?*

Response: This is an important issue and we thank the reviewer for pointing this out. The belt-like structures were flat and long structures with a limited width. As shown in

Fig. 3c in the manuscript, most of these belts appeared to be bent in the x-y plane instead of straight as indicated by the red circles in Fig. 3c, Fig. S51, and Fig. S62b in the original manuscript and supplementary information. In order to clarify these claims, we have rephrased our claim in the manuscript on page 6 from:

“More interestingly, most of these belts appeared to be bent instead of straight, which has been rarely observed.”

to now read

“More interestingly, most of these belts appeared to be bent instead of straight as indicated by the red circles in Fig. 3c, Fig. S51, and Fig. S62b, which has been rarely observed.”

The WAXS spectra of the pure belts were obtained and showed hexagonal columnar packings of the HAT mesogens with a peak of spacing about 3.41 \AA^{-1} in the wide-angle region characteristic of π - π stacking in Fig. S56 in the original supplementary information. Additionally, the belts were measured from AFM images (Fig. S51), for which a height of 7 nm. Combining the experimental results from WAXS and AFM, we concluded that the height of the belts was the height of π - π stacking distance of ten HAT discs and five TNF discs around 5.1 nm. It is worth noting that the theoretical height value was slightly lower than the actual measured height due to the existence of the collapsed coronal P β BA chains.

Comment 4: *What do the authors mean when they talk about lamellar ordering of columnar mesogens (note: columnar refers to a kind of mesophase, not a type of mesogen)?*

Response: We thank the reviewer for pointing this out. As mentioned in the literature paper (Macromolecules 2015, 48, 19, 6768-6780), a diffuse reflection peak in the low angle region of 36.55 \AA (0.172 \AA^{-1}) appeared to be twice the d-value 18.10 \AA (0.347 \AA^{-1}) of (100) reflection of Col $_{ho}$ lattice (Fig. 2b). Such a SAXS signal presumably originated from a polymer main chain and supercylinders constructed from DCS-based four-column bundles of a single polymer chain. The four-column bundles were organized into hexagonal columnar lattice based on side-chain HAT stacking; further, the four-column bundles within the horizontal direction led to lamellar electron density modulation upon HAT stacked columns to further present some extent lamellar correlation.

To avoid confusion, we have changed the phrase “columnar mesogens” to “columnar mesophase” in the manuscript.

Comment 5: *One other point that I believe should be addressed. The authors cite the work of Ringsdorf and acknowledge his work on DA complexes of of TNF and HAT from several decades ago. Nonetheless, I did not feel that the authors have properly detailed the extent to which observations in present work were anticipated by Ringsdorf's extensive studies. In particular, many of the effects of TNF/HAT complexation (shorter distances, higher ordering) were previously noted in non-polymeric systems. This is not to diminish the novelty of some of the key observations*

in the present work, as Ringsdorf's work was not concerned with solution properties nor block copolymers.

Response: This is an important issue and we thank the reviewer for pointing this out. To better illustrate the novelty of this work, we have made an additional statement in the manuscript on page 7 as follows,

“Very similar phenomena were reported in the previous study that LC orderedness could be greatly enhanced by doping the mesogens with small-molecule dopants, allowing the induction of mesophases even in previously non-liquid-crystalline polymers⁴⁸ However, these studies only investigated their self-assembly behaviors in the bulk state, but no solution-state behavior has ever been explored.”

Comment 6: *Finally, I found the repeated use of the term "instantly" to describe the effects of TNF on morphology inappropriate; a more scientific approach to describing timescales is in order.*

Response: This is an important issue. We believe the term "instantly" is applicable to describe the complexation of EDA, when TNF was added to the tetrahydrofuran (THF) solution of PHATMA the colorless solution immediately turned black within 3 seconds, due to the formation of EDA. We only used the term "instantly" to describe the mesogenic ordering, rather than the effects of TNF on morphology in the original manuscript. Now we have replaced it with "very quickly" or "immediately within 3 seconds" to illustrate the process more precisely.

Reviewer 2:

The reviewer makes three suggestions:

Comment 1: *Does the mesogen functionalized block copolymer form any lyotropic liquid crystal phase at higher concentrations than used in the study?*

Response: This is an important issue. In fact, we are not sure whether these block copolymer micelles form any lyotropic liquid crystal phase at concentrations much higher than the current ones. According to literature reports, this should occur at a concentration at least 100 times higher than the one used in the current study (0.1 mg/mL). For example, Zhihong Nie et al. (Sci. Adv. 2018; 4: eaas8829) have systematically examined the phase behaviors of colloidal analogs of bent-core LC molecules and determined the role of rod geometry in the formation of different ordering phases at high concentrations (volume fraction of 30%). That corresponds to at least 3000 times higher than the concentration used in the current study. Based on the findings of that paper and related studies, we tend to believe that if these micelles are fabricated at much higher concentrations, they could possibly form a lyotropic liquid crystal phase. However, these studies would go beyond the scope of the current study and may be covered in future studies.

Comment 2: *The authors mention that "worm-like aggregates were further swollen into tubular structures". How did the authors confirm the tubular nature of the*

aggregates? Experimental data should be provided for this claim i.e., TEM or SEM images. Similarly, they state the tubular structures developed into vesicles. I do not see evidence for the vesicle structures. How did they differentiate between micellar structure and vesicular morphologies?? Electron density map calculations might be of help.

Response: This is an important issue and we are grateful to the reviewer for pointing this out. The different morphologies can be distinguished from their microscopic images. As shown by the TEM image in Fig. 1d in the original manuscript, the worm-like structures appeared to be solid, and the contrast was high. Meanwhile, as shown in Fig. 1e in the original manuscript, the contrast of the middle area of the tubular structures was lower than the other area, suggesting a hollow nature. Additionally, we have performed additional TEM experiments and grayscale analyses (Fig. S21) to confirm the hollow structure of the tubes. The grayscale analyses were performed with the Digital Micrograph software package (Gatan). This is supported by grayscale analyses over selected cross-sectional areas, revealing the hollows of tubular structures.

Additional discussions were also included as:

“The hollow tubular structure was verified with grayscale analysis (Fig. S21).”

Figure S21 (a, c) TEM images and (b, d) the corresponding grayscale analyses of the tubular structures by dispersing the P*t*BA-*b*-PHATMA in 2-PrOH at 80 °C for 1 h and after the solution reached r.t.

Regarding the vesicles, their hollow nature could be verified from their TEM images (Fig. 1f and 1g), showing higher contrast along the side of these structures but lower contrast inside. Secondly, we have performed additional TEM and AFM experiments (Fig. S22). As shown in Fig. S22a, two layers of membranes of the vesicle could be

clearly observed from the broken vesicle. The wrinkles could be visualized from these structures after they were dried for TEM (Fig. 1g) and AFM (Fig. S22b) observations, a typical feature for vesicular structures (Polymer 2020, 207, 122914; J. Am. Chem. Soc. 2011, 133, 16581-16587; Chem. Soc. Rev. 2019, 48, 4019-4035; Science 2002, 297, 967-973).

Additional discussions were also included as:

"The vesicular structures were also confirmed by the observation of wrinkles from their TEM, atomic force microscopic (AFM) images, and also grayscale analysis (Fig. S22, and 23)."

Figure S22 (a) TEM and (b) AFM topographic images of the vesicular structures by dispersing the PtBA-*b*-PHATMA in 2-PrOH at 80 °C for 1 h and cooling down to r.t. within 10 h storage.

We have performed additional TEM grayscale analyses, as shown in Fig. S23. The grayscale analyses were performed with the Digital Micrograph software package (Gatan). This is supported by grayscale analyses over selected cross-sectional areas, revealing the wrinkles of vesicular structures. The wrinkles could appear dark from these structures (Fig. S23), similar to TEM images.

Figure S23 (a) TEM images and (b) the corresponding grayscale analyses of the vesicular structures by dispersing the PtBA-*b*-PHATMA in 2-PrOH at 80 °C for 1 h and cooling down to r.t. within 10 h storage.

Comment 3: *The XRD data in figure 2b seems to indicate a columnar lamellar ordering but the authors claim the occurrence of hexagonal columnar ordering. The exact nature*

of the mesophase should be resolved probably using fast Fourier transform (FFT) of the TEM images which could reveal the underlying ordering in the self-assembled structure.

Response: This is an important issue and we thank the reviewer for pointing this out. Indeed, we could not observe the exact nature of the mesophase *via* fast Fourier transform (FFT) of the TEM images. This was due to the fragile LC packing of the discotic mesogens under high-dose electron beams, which could easily melt the LC phase and destroy the LC packings. Additionally, the existence of a coronal layer caused the visualization of the LC packing inside the micellar core extremely difficult.

Therefore, we had to use wide-angle X-ray scattering (WAXS) to characterize the LC packing inside micellar cores. Our WAXS results fully agreed with that from the literature (Macromolecules 2015, 48, 19, 6768-6780; Polym. Chem. 2017, 8, 3286-3293; J. Polym. Sci. Pol. Chem. 2017, 55, 2544-2553; Angew. Chem. Int. Ed. 2013, 125, 1065-1068). A diffuse reflection peak in the low angle region of 36.55 Å (0.172 Å⁻¹) appeared to be twice the d-value 18.10 Å (0.347 Å⁻¹) of (100) reflection of Col_{h0} lattice from the lamella packing of the columnar mesophase. Therefore, we tend to believe that the discotic mesogens also formed hexagonal columnar packings.

Reviewer 3:

The reviewer makes seven suggestions:

Comment 1: *Though the solution self-assembly of synthetic copolymers seems to perform a variant kinetics as compared to the bulk assembly of the same comparable copolymers referring to Macromolecules 2015, 48, 6768-6780, for TP-based homopolymers and J Polym Sci, Part A: Polym Chem 2017, 55, 2544-2553 for TP-based diblock copolymers.*

Response: This is an important issue and we thank the reviewer for pointing this out. To better illustrate the novelty of this work, we have made an additional statement in the manuscript on page 4 as follows,

“The ordering of the LC phase was obviously slower than in the bulk,^{30, 33} probably due to the plasticizing effect from the surrounding solvent molecules. So, it is possible that the LC ordering effect was more complicated in the morphological transition process due to the higher fluidity in the solution and interactions with solvent molecules.”

Comment 2: *The assembly mechanism change into nucleation-growth mode for the highly doped complexes may be not so surprising, in my opinion, it reflects a transformation from a kinetic controlled process for the pristine TP-based diblock copolymer into nucleation-growth mode of thermodynamic at or close to the equilibrium state which may be ascribed to the adoption of small-molecule dopant TNF facilitating the ease in reaching thermodynamic equilibrium state. Otherwise, if TNF was introduced as side group in side-chain polymer as did by Percec et al. (Nature, 2002, 419, 384-387), it may exhibit quite different kinetic behavior. The situation mentioned above in (1) and (2) is suggested to be incorporated into the discussions for kinetics*

versus thermodynamic processes, and compared to the bulk assembly.

Response: We really thank the reviewer for pointing this out. To illustrate the change of assembly mechanism to nucleation-growth after doping more properly, we have made an additional statement in the manuscript on page 9 as follows,

“LC orderedness of the diblock copolymer could be built very quickly by doping the mesogens with small-molecule dopants, and the morphological transitions were dramatically accelerated. The adoption of small-molecule dopant TNF plausibly facilitated the system to reach a thermodynamic equilibrium state. Therefore, the assembly route was transformed from kinetic-controlled for the pristine diblock copolymer into thermodynamic-favored nucleation-growth mode at or close to the equilibrium state. It could be envisioned that if TNF was introduced as side group in side-chain polymer,⁵⁸ it may exhibit quite different kinetic behavior.”

Comment 3: *“When they were attached as pendant groups for polyacrylates, the organization patterns were mostly retained,^{27, 28} but their LC properties were severely impaired, due to the limited mobility from the polyacrylate backbone.²⁹” This sentence is an untimely and incorrect statement from ref.27 (year 2000), ref.29 (year 1989) with some misunderstanding, not in line with the updated cognition for discotic side-chain liquid crystalline polymers as renewed in ref. 28 and *Macromolecules* 2015, 48, 6768-6780 revealing well-organized columnar superlattices via positive coupling between the polymer backbone and discotic side groups. Actually, TP-based discotic columnar liquid-crystalline polymer semiconducting materials via rational macromolecular engineering showed high charge-carrier mobility, which was attributed to significantly enhanced orientational correlation along polymer backbone (, together with dynamic disorder inhibition and easy columnar alignment, though the restricted rotational and translational movements in side-chain discotic liquid crystalline polymers (*Polym. Chem.* 2017, 8, 3286-3293).*

Response: We are really grateful to the reviewer for highlighting this important issue. In order to clarify these claims, we have added the four papers as references #27, 28, 29, and 30, and rephrased our claim in the manuscript on page 3 from:

“When they were attached as pendant groups for polyacrylates, the organization patterns were mostly retained,^{27, 28} but their LC properties were severely impaired, due to the limited mobility from the polyacrylate backbone.²⁹”

to now read

“When they were attached as pendant groups for polyacrylates, the organization patterns would be mostly retained.²⁷ Meanwhile, their LC properties could be significantly influenced positively or negatively.²⁸⁻³⁰ In the current system, the LC properties were severely impaired, probably due to the introduction of additional rigid methacrylate backbone.”

Comment 4: *A closely related reference (*Polym. Chem.* 2017, 8, 3457-3463) should be additionally cited, which demonstrated self-assembled helical columnar superstructures with selective homochirality from TP-based side-chain discotic liquid*

crystalline polymer doped with chiral acceptor molecules of ester derivatives of TNF to construct electron donor–acceptor (EDA) complexes through CT interactions.

Response: We really appreciate the reviewer for pointing this out. We have cited the reference (Polym. Chem. 2017, 8, 3457-3463) in the manuscript as references #43,

“and significantly enhance the orderedness and thermal stability of the LC phase.”^{42, 43}

Comment 5: *“for which a diameter of 25.6 ± 4.6 nm, and a height of 2.0 ± 0.1 nm were measured...” “with a diameter of 45.1 ± 7.0 nm (Fig. 4a)...” Here, for belt-like morphologies, width may be more suitable than diameter?*

Response: We really appreciate that the reviewer pointed this out. We have changed the word “*diameter*” to “*width*” in the manuscript.

Comment 6: *“At this stage, the orderedness of the disks inclined over time, ...” “With an inclining annealing temperature from 30 °C to 72 °C, the fibrils...” Here, “inclined”, “inclining”, seems to be changed into increased, increasing?*

Response: We really appreciate that the reviewer pointed this out. We have changed the words “*inclined*” to “*increased*” and “*inclining*” to “*increasing*” in the manuscript.

Comment 7: *In the Supporting Information file, in the ¹H NMR spectra of Figure S1-S6, S28, and of Figure S9, S13, S30, the labelled CDCl₃ or CD₂Cl₂ is improper, those peaks are from the residual non-deuterated solvents, deuterated solvents have no ¹H NMR signal! In Figure S27, the caption “(b) then stored storing for 6 months at r. t.” change into “(b) then stored for 6 months at r. t.”*

Response: We appreciate that the reviewer pointed this out. We have changed the phrase “*then stored storing for 6 months at r.t.*” to “*then stored for 6 months at r.t.*” in the manuscript.

As for the NMR peak assignment, it is common to label the solvent peaks like this to indicate which solvent was used for the experiment. Many literature reports (e.g. Macromolecules 2015, 48, 2388-2398; J. Am. Chem. Soc. 2018, 140, 18104-18114) labeled these residual non-deuterated solvent peaks.

We believe that the manuscript has been substantially improved by the reviewers' critiques and hope that our responses to their points are satisfactory.

Yours sincerely,

Prof. Xiaoyu Li

School of Material Science and Engineering

Beijing Institute of Technology

REVIEWERS' COMMENTS

Reviewer #1 (Remarks to the Author):

The authors have adequately addressed my comments/concerns.

Reviewer #2 (Remarks to the Author):

The authors have addressed my concerns in the revised manuscript.

Reviewer #3 (Remarks to the Author):

The revised version is further enhanced with addressing the Reviewers' concerns and adopting most of the revision suggestions or requirements. The most important contribution of this work is to provide and well demonstrate the doping-triggered controllable assembly as a new approach and effective means to expand and deepen the solution-state self-assembly scenario of synthetic block copolymers, which may have highly significant implications for better understanding or mimicking the delicate and elegant assembly behaviors of complex biological systems.

After further complete the following minor revisions, I am very pleased to see its being published in the prestigious top journal of Nature Communications.

(1) ".....DSC characterization and record the results from the first heating scan (Fig. S19)." While in Fig. S19 of the Supporting Information, it was clearly labelled as "2nd heating".

(2) ".....but instead flat bilayer structures were preferred to accommodate the high bending penalty of the mesogenic phase." ".....due to the existence of highly doped diblock copolymers and the formation of a highly ordered mesogenic phase." What does "mesogenic phase" mean here?

(3) "Surprisingly, if the discotic mesogens were doped to form energy-donor/accept complexes,....." Here "energy-donor/accept complexes" should be electron donor-acceptor complexes.

(4) "Topographic image (Fig. 2c) of the thin fibrils were first obtained,....." change into ".....image (Fig. 2c).....was.....".

(5) "The detailed molecular characteristics of the diblock copolymer were included....." Also appeared in Table S1 and S2, "Molecular characteristics of....." Change the "characteristics" into characterization.

(6) ".....such as the UV irradiation-induced morphological switch....., etc."such as....., etc. Please pay attention to the matching grammar problem.

(7) In the Supporting Information, Figure S61, ".....of the undoped fibrils 2-PrOH solution ($r=1$)..." undoped... $r=1$?

(8) In Figure 3, ".....by dispersing the PtBA-b-PHATMA with various r -values..." Also appeared in the Supporting Information, Figure S41, S45, and Figure S50, ".....by dispersing the PtBA-b-PHATMA ($r = 0.4$)..." Figure S52, ".....by dispersing the PtBA-b-PHATMA ($r = 0.9$)..." It is obviously that "doped" should

be inserted to read as "...by dispersing the doped PtBA-b-PHATMA....."

(9) In the Supporting Information file, in the ^1H NMR spectra of Figure S1-S6, S28, and of Figure S9, S13, S30, the labelled CDCl_3 or CD_2Cl_2 is improper, those peaks are from the residual non-deuterated solvents, deuterated solvents have no ^1H NMR signal! (Answer by the authors: As for the NMR peak assignment, it is common to label the solvent peaks like this to indicate which solvent was used for the experiment.)

I do not think it is reasonable to adhere to the confused labeling based on some paper many years ago, please refer to the strictly normalized right labeling, such as in *Macromolecules* 2019, 52, 6913-6926 and many others.

Re: Manuscript number: NCOMMS-23-40393B

Title: "Convoluting Micellar Morphological Transitions Driven by Tailorable Mesogenic Ordering Effect from Discotic Mesogen-Containing Block Copolymer"

Comment 1: *".....DSC characterization and record the results from the first heating scan (Fig. S19)." While in Fig. S19 of the Supporting Information, it was clearly labelled as "2nd heating".*

Response: We have removed the wrong labelling of "2 nd heating" in Fig. S19. It is now correctly labelled as "1 st heating".

Comment 2: *".....but instead flat bilayer structures were preferred to accommodate the high bending penalty of the mesogenic phase." ".....due to the existence of highly doped diblock copolymers and the formation of a highly ordered mesogenic phase." What does "mesogenic phase" mean here?*

Response: We have revised the sentence from

"to accommodate the high bending penalty of the mesogenic phase."

into

"to accommodate the high bending penalty of the mesogens."

and revised the sentence from

"and the formation of a highly ordered mesogenic phase."

into

"and the formation of a highly ordered LC phase."

Comment 3: *"Surprisingly, if the discotic mesogens were doped to form energy-donor/accept complexes,....." Here "energy-donor/accept complexes" should be electron donor-acceptor complexes.*

Response: We have changed "energy donor/acceptor complexes" to "electron donor-acceptor complexes".

Comment 4: *"Topographic image (Fig. 2c) of the thin fibrils were first obtained,....." change into ".....image (Fig. 2c).....was.....".*

Response: We have revised the manuscript accordingly.

Comment 5: *"The detailed molecular characteristics of the diblock copolymer were included....." Also appeared in Table S1 and S2, "Molecular characteristics of....." Change the "characteristics" into characterization.*

Response: We have revised the manuscript accordingly.

Comment 6: *".....such as the UV irradiation-induced morphological switch....., etc."such as....., etc. Please pay attention to the matching grammar problem.*

Response: We have removed "etc".

Comment 7: *In the Supporting Information, Figure S61, “.....of the undoped fibrils 2-PrOH solution (r=1)...” undoped...r=1?*

Response: We have removed “(r=1)”.

Comment 8: *In Figure 3, “.....by dispersing the PtBA-b-PHATMA with various r-values...” Also appeared in the Supporting Information, Figure S41, S45, and Figure S50, “.....by dispersing the PtBA-b-PHATMA (r = 0.4)...” Figure S52, “.....by dispersing the PtBA-b-PHATMA (r = 0.9)...” It is obviously that “doped” should be inserted to read as “...by dispersing the doped PtBA-b-PHATMA.....”*

Response: We have revised the manuscript accordingly.

Comment 9: *In the Supporting Information file, in the ¹H NMR spectra of Figure S1-S6, S28, and of Figure S9, S13, S30, the labelled CDCl₃ or CD₂Cl₂ is improper, those peaks are from the residual non-deuterated solvents, deuterated solvents have no ¹H NMR signal! (Answer by the authors: As for the NMR peak assignment, it is common to label the solvent peaks like this to indicate which solvent was used for the experiment.)*

I do not think it is reasonable to adhere to the confused labeling based on some paper many years ago, please refer to the strictly normalized right labeling, such as in Macromolecules 2019, 52, 6913-6926 and many others.

Response: We have revised the manuscript accordingly.